# Circulating Tumor Cells in Metastatic Breast Cancer: Clinical Applications and Future Possibilities

**Maggie Banys-Paluchowski [1],\*, Florian Reinhardt [2] and Tanja Fehm [2]**

[1]    Department of Gynecology, Asklepios Klinik Hamburg-Barmbek, 22307 Hamburg, Germany
[2]    Department of Obstetrics and Gynecology, University of Düsseldorf, 40225 Düsseldorf, Germany;
        florian.reinhardt@med.uni-duesseldorf.de (F.R.); tanja.fehm@med.uni-duesseldorf.de (T.F.)
\*    Correspondence: m.banys@outlook.com

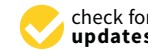

**Featured Application: Circulating tumor cells (CTCs) have become one of the key research fields in translational oncology. The molecular characterization of CTCs offers a unique insight into genotypic and phenotypic dynamics of metastatic disease.**

**Abstract:** Circulating tumor cells (CTCs) have gained importance as an emerging biomarker in solid tumors in the last two decades. Several detection assays have been introduced by various study groups, with EpCAM-based CellSearch system being the most widely used and standardized technique. In breast cancer, detection of CTCs correlates with clinical outcome in early and metastatic settings. CTC persistence beyond first cycle of palliative chemotherapy indicates poor response to treatment in metastatic situation. Beyond prognostication and therapy monitoring, CTC counts can guide treatment decisions in hormone receptor positive HER2-negative metastatic breast cancer. Furthermore, CTC-based therapy interventions are currently under investigation in clinical trials. In this review, we focus on the current state of knowledge and possible clinical applications of CTC diagnostics in patients with metastatic breast cancer.

**Keywords:** breast cancer; liquid biopsy; circulating tumor cell; survival; targeted therapy

## 1. Introduction

The use of blood-based diagnostics in cancer patients is referred to as 'liquid biopsy' and has become a major focus of translational research in the last decades. In breast cancer, the detection of both circulating tumor cells (CTCs) and circulating tumor DNA (ctDNA) have been extensively investigated in clinical trials [1]. It is widely accepted that CTCs serve as precursors of metastatic disease. As such, they are commonly referred to as minimal residual disease (MRD) in patients with early breast cancer treated by curative surgery and/or systemic therapy. Clinical studies and animal-based models have shown that hematogenous tumor cell dissemination occurs early in the course of disease and may already be detected in patients with lesions considered preinvasive by conventional histopathology [2,3].

In metastatic breast cancer, most applications currently under study in clinical trials are based on the assumption that tumor cells detected in the blood stream reflect the dominant cell populations of distant metastatic sites. Therefore, liquid biopsy may provide valuable insight into the current state of disease and spare at least some patients an invasive procedure for metastasis biopsy. In addition, since serial biopsies of distant organs are not feasible, non-invasive evaluation of CTCs may potentially serve as a 'real-time' biopsy and inform treatment decisions.

In this review, we discuss the current data on the clinical role of CTCs in patients with metastatic breast cancer and potential new applications of CTC diagnostics in the future.

## 2. Prognostic Relevance of CTC Counts

While most studies conducted in early breast cancer setting stratified patients based on the presence of at least one CTC (i.e., into a CTC-negative and a CTC-positive subgroup), this approach is less feasible in metastatic disease. In general, tumor cell counts are significantly higher in patients with advanced cancer. Therefore, the most widely used cut-off value in metastatic breast cancer is 5 CTCs per 7.5 mL peripheral blood using the only FDA approved detection system (CellSearch) [4].

The first report on the prognostic value of CellSearch-based CTC detection in metastatic breast cancer was published in 2004 by Cristofanilli et al. from the M.D. Anderson Cancer Center [5]. These results have been since confirmed in a large meta-analysis of data from 18 centers [6]. In this study, blood samples from 2,436 patients were analyzed using CellSearch at time of the first diagnosis of metastatic disease or progression. Elevated CTC counts (≥5 CTCs) were found in 47% of patients. After a median follow up of 14.9 months, CTC numbers were associated with overall survival (OS; 16 vs. 36.3 months in CTC-high and CTC-low groups, respectively). This observation was independent of the tumor subtype and the localization of metastatic disease (OS in patients with visceral metastasis: 13.2 vs. 29.9 months and in patients with bone-only disease: 23.8 vs. 46.9 months, respectively).

## 3. Therapy Monitoring Using CTC Counts

Large trials have already proven the prognostic relevance of CTCs in metastatic breast cancer. Furthermore, there is evidence that monitoring CTC dynamics during the course of therapy can predict treatment response [7]. In most breast cancer patients, CTC numbers declined rapidly under systemic therapies compared to those before therapy started [8]. In the study by Martin et al., CTCs were measured in patients receiving palliative chemotherapy. In the multivariate analysis, response to therapy and survival was associated with CTC counts after the first cycle of therapy [9]. Respectively, monitoring of CTC numbers might be able to predict sooner than conventional radiological imaging whether patients will benefit from therapy. Nevertheless, which therapeutic strategy for non-responders should be offered remains unclear.

In the randomized phase III SWOG S0500 trial (NCT00382018), CTC levels were determined in 595 metastasized patients before and during first-line chemotherapy [10]. Out of this study cohort, 123 patients had increased CTC numbers prior to chemotherapy as well as after the first cycle. Patients were randomized either to maintain therapy or switch to another chemotherapy regime. Treatment change based on CTC persistence did not improve PFS or OS in these patients (PFS 4.6 vs. 3.5 months, HR = 0.92; 95%CI = 0.64–1.32; OS 1.5 vs. 10.7 months, HR = 1.00, 95%CI = 0.69–1.47). It is possible that these patients might represent a chemo-resistant population independent of the cytotoxic drug used that requires targeted or alternative treatment approaches.

Moreover, the currently ongoing multicenter randomized Phase III CirCe01 study (NCT01349842) monitors CTC dynamics for therapy guidance. Besides clinical tests and radiologic imaging, CTC counts are assessed for therapy response in CTC-positive patients after two lines of therapy. Patients without a significant decrease in CTC counts after the first cycle of new chemotherapy are switched to an alternative regime, which is also be monitored by CTCs. To date, there are no results published. Both large trials attempt to demonstrate that patients with persistently elevated CTCs counts under chemotherapies should be early switched to targeted or alternative therapies to avoid inefficient and toxic therapies.

## 4. Therapy Selection Based on CTCs

So far, treatment decisions in metastatic breast cancer are based on the histologically verified predictive features of the disease. This makes invasive and potentially harmful biopsies of distant metastatic sites, such as lung or liver, necessary in order to recommend adequate systemic therapy. In patients where a metastatic biopsy is not possible (e.g., due to an untypical localization or patient's condition), the choice of therapeutic strategy is based on the properties of the primary tumor, which in

some cases had been removed several years ago. However, the pheno-and genotype of tumor cells may change during the course of disease. According to a large meta-analysis, a switch of receptor status can be observed in up to one third of metastatic patients (Table 1, [11]). Therefore, current guidelines recommend a re-evaluation of predictive features of tumor cells via a metastatic biopsy. However, this approach does not address a potential heterogeneity between different metastatic sites (or even different parts of the same metastasis). Furthermore, serial metastatic biopsies are hardly feasible.

**Table 1.** Discrepancy between receptor status of the primary tumor and the metastasis (modified after: [11])

|  | ER | PR | HER2 |
|---|---|---|---|
| Change of receptor status (total) | 19.3% | 30.9% | 10.3% |
| Conversion to negative in case of initially positive receptor status ($+ \rightarrow -$) | 22.5% | 49.4% | 21.3% |
| Conversion to positive in case of initially negative receptor status ($- \rightarrow +$) | 21.5% | 15.9% | 9.5% |

### 4.1. Therapy Interventions Based on CTC Counts

Given that CTCs seem to reflect the features of dominant metastatic cell populations, evaluation of their counts as well as geno-and phenotypes may provide insight into current dynamics of the disease and inform treatment decisions. The first positive study on CTC-based therapy interventions was the STIC CTC trial initiated in France by the Institut Curie [12]. A total of 778 patients with metastatic hormone receptor (HR) positive HER2-negative breast cancer received first-line therapy within this phase III study. CTC numbers were assessed using CellSearch before start of treatment and were elevated in 27% of patients. Patients were randomized into two groups. In the standard arm, they received therapy of physician's choice (chemotherapy or endocrine therapy). In the CTC-arm, the therapy was guided by CTC counts: patients with ≥5 CTCs received chemotherapy, patients with <5 CTCs endocrine therapy. After a median follow up of 30 months, the progression-free and overall survival were similar in both groups, i.e., CTC counts stratified patients into low-risk and high-risk groups about the same as an experienced oncologist. Moreover, in case of discrepant risk estimation (i.e., clinically low-risk patients with ≥5 CTCs or clinically high-risk patients with <5 CTCs) chemotherapy led to significantly longer overall survival than endocrine therapy. The results of the STIC CTC trial have been extensively discussed after their publication at the San Antonio Breast Cancer Symposium 2018 [12]. While the study has demonstrated the value of liquid biopsy-based treatment selection, it remains open how it might be implemented in the clinical routine. Since the trial has begun before the approval of CDK4/6 inhibitors, endocrine-based combination therapy has not been incorporated in the study design. However, currently most patients with HR+ HER2- metastatic breast cancer receive CDK4/6-based therapy and it is unclear how CTC diagnostics may guide treatment decisions in this setting.

### 4.2. Therapy Interventions Based on Molecular Characteristics of CTCs

Beyond simple enumeration of CTCs, assessment of their predictive features for therapy interventions is the focus of other trials. The aim of the Circe T-DM1 trial was the evaluation of HER2-directed treatment for patients with histologically HER2-negative metastatic disease and HER2-amplified CTCs [13]. To date, such patients receive no HER2-targeted treatment because the indication is based on the histological receptor status of the primary tumor and/or metastasis. Blood samples from 154 heavily pretreated patients were evaluated in the CirCe T-DM1 study. The HER2 status of the detected CTCs was assessed using fluorescence in situ hybridization (FISH). At least one HER2-amplified CTC was detected in 14 patients. However, the response to targeted therapy with the antibody-drug-conjugate trastuzumab-emtansin was low. Among 11 patients treated within this trial, only one reached partial response and four had achieved stable disease. One possible explanation for the negative result of the study was the very low incidence of HER2-positive CTCs. The majority of patients harbored only one HER2-amplified tumor cell alongside several HER2-negative CTCs. Indeed,

in the group of patients with at least one HER2-positive CTC, only 4.4% of CTCs evaluated using FISH were HER2-amplified. Therefore, the clinical relevance of this minor subpopulation seems limited.

Another approach is currently under investigation in the German DETECT studies, the largest research program on CTC-based therapy interventions worldwide (www.detect-studien.de) [1]. Here, patients with histologically HER2-negative disease are screened for HER2-positive CTCs (Figure 1). Within the phase III trial DETECT III (NCT01619111), they receive standard treatment +/− HER2-directed therapy with lapatinib. Patients with HER2-negative CTCs may participate in the DETECT IV trial (NCT02035813), which investigates the dynamics of CTCs under treatment with ribociclib or eribulin. Another of the DETECT studies, the phase III trial DETECT V is open for patients with HER2-positive disease (NCT02344472). These trials are still recruiting. The first results are expected in 2021.

Table 2. shows the current state of knowledge on the clinical role of CTCs in metastatic breast cancer.

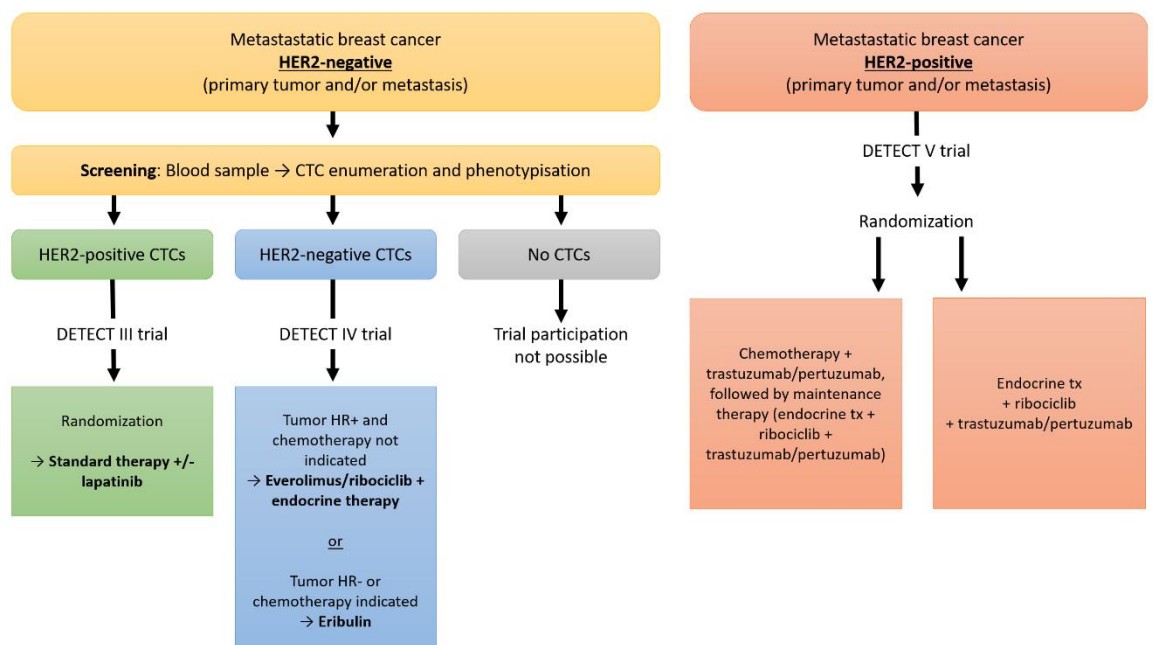

**Figure 1.** Design of the DETECT trials.

**Table 2.** Clinical relevance of CTCs in metastatic breast cancer

| Breast Cancer Stage IV | |
|---|---|
| Prognosis assessment | • Detection of ≥5 CTCs per 7.5 mL blood (CellSearch) at time of diagnosis of distant metastasis or progression is strongly associated with worse survival<br>• Most important references: [5,6,10,14–19] |
| Therapy monitoring | • Persistence of CTCs after first cycle of chemotherapy correlated with response to treatment; patients do not benefit from a switch to another chemotherapy regimen (SWOG S0500 trial)<br>• Most important references: [9,10,20–23] |
| CTC-based therapy interventions | • HR+ HER2- breast cancer: CTC-based choice of first-line therapy is not inferior to therapy of physician's choice (STIC CTC trial); implementation in the clinical routine unclear due to approval of endocrine-based combination therapy with CDK4/6 inhibitors<br>• HER2- breast cancer: Patients do not benefit from T-DM1 in case of histologically HER2-negative disease and HER2-amplified CTCs (Circe T-DM1 trial)<br>• Further studies ongoing (e.g., DETECT trials)<br>• Most important references: [12,13,20,24,25] |

## 5. Future Possibilities

Treatment of metastatic breast cancer patients has become increasingly individualized in recent years. Optimization of personalized treatment regimens in real-time to effectively treat breast cancer patients is urgently needed. CTCs have the potential to base personalized medicine to the next stage. The prognostic relevance of CTCs has already been shown for metastatic breast cancer. There is evidence that monitoring of CTC dynamics during the course of therapy can predict treatment response. Thus, CTCs as suitable tool for prognosis and treatment monitoring have been also partly included into clinical guidelines. To bring CTCs into the realm of routine clinical decision making, large clinical trials are currently investigating their clinical role for treatment optimizations with regard to extensions, modifications, or abandonments of treatment regimes.

**Author Contributions:** Writing—original draft preparation and review/editing: M.B.-P., F.R., and T.F. All authors have read and agreed to the published version of the manuscript.

**Funding:** This research received no external funding.

**Conflicts of Interest:** The authors declare no conflict of interest.

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
