# Peer review of "Circulating Tumor Cells in Metastatic Breast Cancer: Clinical Applications and Future Possibilities"

_applsci, doi:10.3390/app10093311_

Round 1
Reviewer 1 Report
The topic regarding the study of CTCs in breast cancer has been reviewed many times in recent years. The current review article is expected to bring an updated version with latest studies.
- As a review article, it would be suggested to add a table to summarize all the related references to the application of CTC enumeration in MBC prognosis or/and diagnosis.
- Please re-organize the structure of the text to make it clearer, such as CTC counts in clinical application and molecular analysis of CTCs in the application, etc.
- The format of the tables and the resolution of the figure 1 need to be improved to a GREAT extent.
Reviewer 2 Report
I have some minor changes.
line 53 ..cancer was published in 2004...
line 65 .numbers declined rapidly..
line 66 ..before therapy started
line 78..It is possible that these..
line 92 In patients where a ..
line 94 ..which in some cases had been..
line 115 ..groups about the same as an..
lines 136-138..this is a run-on sentence that needs to be broken down
Round 2
Reviewer 1 Report
Further format editing is recommended, such as to avoid the double spread of a table.